# The Epigenetic Machinery and Energy Expenditure: A Network to Be Revealed

**DOI:** 10.3390/genes16010104

**Published:** 2025-01-19

**Authors:** Elisabetta Prada, Giulia Bruna Marchetti, Denise Pires Marafon, Alessandra Mazzocchi, Giulietta Scuvera, Lidia Pezzani, Carlo Agostoni, Donatella Milani

**Affiliations:** 1Azienda Socio Sanitaria Territoriale Lariana, 22100 Como, Italy; elisabetta.prada@asst-lariana.it; 2Fondazione IRCSS Ca’ Granda Ospedale Maggiore Policlinico, 20122 Milano, Italy; alessandra.mazzocchi@unimi.it (A.M.); giulietta.scuvera@policlinico.mi.it (G.S.); lidia.pezzani@policlinico.mi.it (L.P.); carlo.agostoni@unimi.it (C.A.); donatella.milani@policlinico.mi.it (D.M.); 3Department of Public Health and Infectious Diseases, Specialization School in Medical Statistics and Biometry, Università Sapienza di Roma, 00185 Roma, Italy; 4Laboratorio di Genetica Medica, ASST Papa Giovanni XXIII, 24127 Bergamo, Italy; 5Department of Clinical and Community Sciences, Università degli Studi di Milano, 20133 Milano, Italy

**Keywords:** Rubinstein–Taybi, sotos, chromatinopathies, rest energy expenditure, metabolism

## Abstract

Mendelian disorders of the epigenetic machinery (MDEMs) include a large number of conditions caused by defective activity of a member of the epigenetic machinery. MDEMs are characterized by multiple congenital abnormalities, intellectual disability and abnormal growth. that can be variably up- or down-regulated. **Background/Objectives**: In several MDEMs, a predisposition to metabolic syndrome and obesity since childhood has been reported. **Methods**: To investigate the metabolic bases of this abnormal growth, we collected physical data from a heterogeneous pool of 38 patients affected by MDEMs. Thirty-five patients performed indirect calorimetry (as a measure of resting energy expenditure, REE) and blood tests to monitor plasmatic nutritional parameters. **Conclusions**: Although limited by a small-sized and heterogeneous sample, our study demonstrates a linear correlation between REE and physical parameters, OFC, height and weight, and observed a slight imbalance on several plasmatic spies of metabolic syndrome predisposition. Furthermore, we demonstrated a significantly higher REE in Sotos Syndrome type 1 patients compared to the controls, which resulted independent from height, suggesting that impaired metabolism in these patients may go beyond overgrowth.

## 1. Introduction

Thanks to the complex wheels of the Epigenetic Machinery (EM), our cells achieve fine-tuning and specific regulation of gene expression. Main Machinery actors include writers and erasers which, respectively, upload and remove marks that switch chromatin state from open to closed and vice versa (e.g., through CpGs methylation and histone tails acetylation), readers, able to recognize these marks and to recruit transcription complexes, and remodelers that modulate the access of ATP-dependent multiple domain protein complexes to DNA [1,2]. The key role played by EM in human development is underscored by the wide spectrum of conditions associated with germline defects in its proteins. To date, 148 genes participating in EM have been causally related to at least 1 of 179 MDEMs (Mendelian disorders of the EM) [3]. While individually rare, MDEMs are increasingly numerous and collectively represent one main cause of neurodevelopmental disorders and growth dysregulation, with both over- and under-growth presentations [4].

Somatic growth is a dynamic process strongly braided to energy metabolism. Total energy expenditure (TEE) comprises thermogenesis, physical activity and basal metabolic rate (BMR), which accounts for 40–70% of TEE and represents the calories required to preserve the body’s homeostasis [5]. Several factors can affect BMR: height and weight, body surface area and composition, age, sex, ethnicity, temperature, diet and hormonal status. In clinical practice, BMR can be calculated using indirect calorimetry, a measure of resting energy expenditure (REE), which empirically exceeds BMR by approximately 10% [5]. Several studies have already demonstrated impairment in BMR in other epigenetic disorders, particularly in imprinting defects (e.g., in Prader–Willi Syndrome [6]). BMR represents the most reliable index of energy expenditure in these conditions, which may exhibit muscle hypotonia and abnormal eating behaviors, affecting physical activity and thermogenesis, respectively [4]. To investigate the potential link between energy expenditure, growth abnormalities and EM defects, we performed indirect calorimetry in a cohort of individuals with MDEMs and investigated its possible connection with anthropometric measurements (weight, height and head circumference) and nutritional indices. Understanding this relationship has the potential not only to improve the diagnosis and management of MDEMs but also to shed light on the pathophysiology of growth abnormalities in these disorders, possibly paving the way for novel therapeutic approaches.

## 2. Materials and Methods

This study was performed at the Paediatric Unit of Fondazione IRCCS Ca’ Granda Ospedale Maggiore Policlinico, Milano. Italy. It was approved by the Ethics Committee and conducted according to Good Clinical Practice and Helsinki Declaration. Patients aged 0–20 years with a genetically confirmed diagnosis of MDEM were enrolled. Clinical evaluation, indirect calorimetry and blood tests were performed on the whole cohort.

Clinical evaluation: Medical records of each patient reporting family history, prenatal and neonatal course, psychomotor development, any congenital malformations and genetic tests were collected to confirm MDEM diagnosis. At physical examination, we measured weight (W), height (H), occipitofruntal circumference (OFC), body mass index (BMI) and converted the values into standard deviation (SD) using the WHO Anthro software (V.3.2.2) for children up to 5 years old and WHO Anthro Plus software (V.1.0.4) for patients older than 5. To unbiasedly compare patients with different MDEMs, we chose not to use syndromes’ specific growth charts, where available.

Indirect calorimetry: All enrolled patients underwent indirect calorimetry with Sensor-Medics Vmax SPECTRA in the morning after overnight fasting; each subject had to breathe under a canopy mask for 20–30 min while he was awake, relaxing quietly. Indirect calorimetry allowed the evaluation of REE, subsequently compared with age- and sex-matched controls. The test failed in three patients (2 RSTS1 and 1 SS1) due to poor compliance.

Blood tests: Blood tests were performed once to value: protein metabolism (prealbumin, leucine); carbohydrate metabolism (glycemia, insulin); fat metabolism (high density lipoprotein, low density lipoprotein, apolipoprotein A, apolipoprotein A1, apolipoprotein A2, apolipoprotein B).

Statistical analysis: According to sample size, we used non-parametric tests to determinate the frequencies (number and percentile) for qualitative variables and the median, the first (Q1) and the third (Q3) quartiles for quantitative ones. Using the Mann–Whitney test, the cases were compared to the controls. A *p*-value < 0.05 was considered statistically significant. Moreover, we calculated the Spearman correlation coefficients among the variables (rho < 0.4: low correlation; rho >= 0.4 e < 0.7: moderate correlation; rho >= 0.7: high correlation). The statistical analysis has been assessed by STATA 15.1 software.

## 3. Results

### 3.1. Description of the Cohort

We enrolled 38 subjects of European ancestry diagnosed with a MDEM, namely 11 individuals with Rubinstein–Taybi syndrome type 1 (RSTS1; OMIM #180849), 10 with Sotos syndrome type 1 (SS1; OMIM #117550), 5 with Kleefstra syndrome type 1 (KLEFS1; OMIM #610253), 3 with Coffin-Siris syndrome type 1 (CSS1; OMIM #135900), 3 with Wiedemann–Steiner syndrome (WDSTS; OMIM #605130), 2 with Smith–Magenis syndrome (SMS; OMIM #182290) and only 1 patient for each of the following conditions: Weaver syndrome (WVS; OMIM #277590), Kabuki syndrome type 1 (KMS1; OMIM #147920), Koolen de Vries syndrome (KDVS; OMIM #610443) and Floating–Harbor syndrome (FLHS; OMIM #136140). The genetic defect of each patient is detailed in Appendix A. Overall, cohort median age was 7.8 years (Q1–Q3 6.4–9.8; min-max 0.9–19) (see Figure 1), with an equal sex ratio (17 male and 21 female, 44.7% and 55.3%, respectively).

### 3.2. Physical, Metabolic and Plasma Nutritional Data

According to growth parameters collected through physical examination and to compare more homogeneous cohorts, we divided patients into two subgroups, those exhibiting growth failure, Group A (including RSTS1, WDSTS, KMS1, KLEFS1, KDVS, SMS, CSS1 and FLHS) [7,8,9,10,11,12,13,14], and those displaying an overgrowth, Group B, namely SS1 and WVS [15,16]. Within the first group, two trends were identified: RSTS1, KLEFS1 and SMS usually show prenatal, neonatal or infancy growth delay and have been overweight since childhood; on the other hand, WDSTS, KMS1, KDVS, CSS1 and FHLS steadily display both weight and height at or under the lower centiles during the prenatal and postnatal age. At blood tests, some parameters mildly deviate from normal ranges in almost all patients: in particular, leucine levels appeared lightly increased (median 110 microMol/L; Q1–Q3 95–122 microMol/L; min–max 80–159 microMol/L) [normal values 75–107 microMol/L], prealbumin was slightly reduced (median 18 microg/dL; Q1–Q3 16–19 microg/dL; min–max 11–23) [normal values 20–40 microg/dL] as well as HDL levels (median 54 mg/dL; Q1–Q3 46–60 mg/dL; min–max 26–78 mg/dL) [normal values >65 mg/dL].

A detailed report of all these parameters can be found in Table 1.

### 3.3. Statistical Analysis

In the overall cohort (N = 35), REE resulted in having a statistically significant and linear correlation to growth parameters with strongest association assessed for OFC (*p* = 0.001, rho = 0.793), followed by weight (*p* = 0.002, rho = 0.758) and height (*p* = 0.005, rho = 0.701) (see Figure 2).

Analyzing potential gender correlations of these associations, we obtained moderate to high correlation values, stronger in males (N = 16, weight rho = 0.78; height rho = 0.68; OFC rho = 0.74) than in females (N = 19, weight rho = 0.65; height rho = 0.52; OFC rho = 0.72).

The correlation analysis between REE and biochemical variables did not reveal any significative association (see Appendix A). We then compared the REE of group A and B with 35 age- and sex-matched controls: no significant difference was observed for Group A (*p* = 0.112). To reduce population heterogeneity and to reach a significative difference, we compared REE of RSTS1 patients versus matched controls and came close to our goal (*p* = 0.053). Notably, the overgrowth group presented a significantly higher REE compared to the control group (*p* = 0.023), independent from height (p=0.0039). The difference was confirmed to be significative of the homogenous cohort of SS1 patients (see Table 2). Focusing on the most represented syndromes among our two subgroups, we compared individuals with RSTS1 and SS1 according to physical, metabolic and plasma nutritional data. These analyses confirmed a statistically significant difference for weight (*p* = 0.016), height (*p* < 0.0001), OFC (*p* < 0.0001), REE (*p* < 0.0001). Also, they demonstrated significative lower levels of insulin (*p* = 0.037) and higher HDL values (*p* = 0.033) in SS1 versus RSTS1 (see Table 3).

## 4. Discussion

MDEMs are caused by defects in epigenetic regulators and characterized by multisystemic involvement and growth dysregulation that, both in excess and in defect, represents a key feature of these disorders [4]. Growth trends can be affected by multiple intrinsic and exogenous factors, including genetic burden, BMR, hormone status, food intake, sucking and swallowing difficulties, poor oro-motor coordination and gastrointestinal disorders [17].

In our cohort, as expected [5,16,17], anthropometric measurements and REE values were proportionally correlated (see Figure 2). Thus, we divided our cohort into two groups based on growth trend. No significant difference was observed for conditions with defective growth (Group A), nor for patients with RSTS1 (see Table 2), demonstrating that small growth MDEMs do not exhibit increased or reduced energy expenditure compared to healthy population, despite their height, age and sex. Still, these results may be biased by small sample size and cohort heterogeneity: genes related to Group A conditions play scattered roles in epigenetic modulation (CREBBP, KMT2A, KMT6A, EHMT1, KANSL1, RAI1, ARID1B and SRCAP) that might result in divergent growth patterns [7,8,9,10,11,12,13,14]. On the other side, the overgrowth group, more homogeneous with 10 SS1 and only 1 WVS, presented a significantly higher REE when compared with age-, sex- and also height-matched controls (see Table 2), confirming the burden of excessive growth on energetic metabolism [5].

Since higher REE values have been shown to be independent of height in SS1 (see Table 2), it can be hypothesized that the caloric metabolism of this condition may be related to unexplored phenomena that go beyond overgrowth.

It will be essential to extend the present study to larger cohorts of MDEMs and to include further data on body composition (free fat mass vs. lean mass), food intake and physical activity in any future analyses aimed at better delineating the metabolism of MDEMs.

Blood tests analyses revealed no significative differences compared to the controls (see Table 1). Still, in the whole cohort lower levels of prealbumin and HDL were observed, together with slightly higher levels of leucine. Leucine is an essential branched-chain amino acid (BCAA) whose excess can be observed in catabolic states (e.g., augmented muscle degradation) [18]. These observations are in line with the known predisposition of several MDEMs to develop insulin resistance and metabolic syndrome in adulthood [19].

The comparison of physical, metabolic and plasma nutritional data between RSTS1 and SS1 revealed significantly lower insulin levels in SS1 (*p* = 0.037), as well as significantly reduced levels of HDL and ApoA2 in RSTS1 (Table 3). Insulin, the main hypoglycemic hormone, plays an important anabolic effect and stimulates protein synthesis through mTORC1 signaling pathway activation [20]. Assuming that, it would be interesting to compare metabolism of SS1 patients to those of overgrowth conditions determined by aberrant up-regulation of mTOR [21]. Also, the low insulin levels in SS1 could be the tip of a sunken impaired anabolic state in this condition and might at least partially explain the augmented REE observed in our cohort. On the other hand, reduced values of HDL in RSTS1 further confirm the predisposition of these patients to metabolic syndrome [19].

The main limitations to our study are represented by the small sample size, due to rarity of investigated conditions, and the lack of data on body composition, lifestyle and hormonal status. Furthermore, the homogeneous European ancestry of the present population may represent a limitation of our study which should be overcome by future studies investigating REE. The assessment of more and more precise data per condition will pave the way for the delineation of genotype–phenotype correlations under a metabolic point of view.

## 5. Conclusions

In conclusion, this is the first description of energy expenditure in MDEMs. We demonstrated a significatively elevated REE in overgrowth conditions and no metabolic expenditure perturbations in those with delayed growth. Blood parameters in the whole cohort confirmed the presence of abnormalities in nutritional values predictive of metabolic syndrome, namely HDL, insulin and leucine. Finally, for SS1 we hypothesize the presence of an imbalanced anabolic activity, testified by low insulin levels and by high energy metabolism, that will need further insights. Future studies on larger cohorts of patients will help to deeper explore these aspects and, hopefully, to set up new therapeutic approaches to growth defects in MDEMs.

## Figures and Tables

**Figure 1 genes-16-00104-f001:**
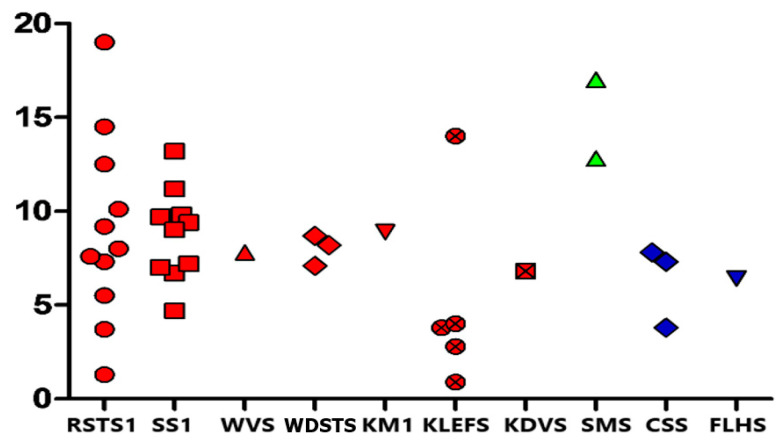
Age distribution of the whole cohort. On Y axe are reported ages, on X axe MDEMs diagnosis; each point represents a patient.

**Figure 2 genes-16-00104-f002:**
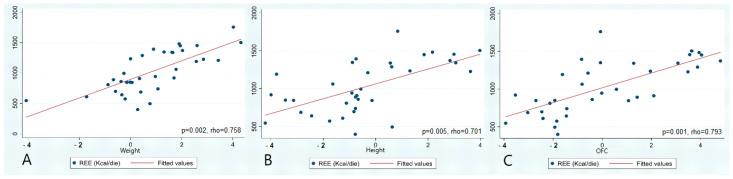
In the overall cohort, a linear correlation was found between REE values and weight (**A**), height (**B**) and OFC (**C**), expressed as standard deviation. P and rho values for each anthropometric parameter are also reported.

**Table 1 genes-16-00104-t001:** Anthropometric parameters and biochemical values of the whole cohort. Values out of ranges are reported in bold font. Abbreviations: BMI = body mass index; CSS: Coffin Siris Syndrome; F = female; FLHS: Floating-Harbor syndrome; Glyc. = Glycaemia; H = height; KDV: Koolen de Vries syndrome; KLEFS1: Kleefstra syndrome type 1; KMS1: Kabuki Syndrome type 1; M = male; OFC = occipitofrontal circumference; Q1 = first quartile; Q3 = third quartile; Prealb. = Prealbumin; REE = rest energy expenditure; RSTS: Rubinstein–Taybi Syndrome; SD = standard deviation; SMS: Smith–Magenis Syndrome; SS1: Sotos Syndrome type 1; W = weight; y = years; WDSTS: Wiedemann–Steiner syndrome; WVS: Weaver syndrome.

	Range and Unit of Measurement	RSTS1 (n = 11)		SS1 (n = 10)		WVS (n = 1)	WDSTS (n = 3)		KMS1 (n = 1)	KLEFS1 (n = 5)		KDV (n = 1)	SMS (n = 2)		CSS1 (n = 3)		FLHS (n = 1)
		Mean	Q1; Q3	Mean	Q1; Q3		Mean	Q1; Q3		Mean	Q1; Q3		Mean	Q1; Q3	Mean	Q1; Q3	
Age	y	8.0	5.5; 12.5	9.2	7.0; 9.8	7.8	8.2	7.1–8.7	8.9	3.8	0.9–6.8	6.8	14.9	12.8–17.0	7.3	3.8–7.8	6.4
Sex		5 M. 6 F		5 M. 5 F		M	2 M. 1 F		F	3 M. 2 F		F	2 F		1 M. 2 F		F
W	SD	−0.02	−0.34; +1.70	+1.77	+0.62; +2.03	+4.31	−0.59	−1.72; +1.31	−0.88	+0.90	+0.76; +1.07	−0.26	+2.20	+0.97; +3.43	+0.06	−0.67; +0.35	−4.08
H	SD	−2.86	−3.79; −1.64	+2.01	+0.96; +2.99	+3.98	−0.89	−1.26; −0.83	−1.12	−0.75	−0.77; +0.63	−0.57	−0.60	−0.91; −0.30	−0.72	−0.77; −0.14	−4.22
OFC	SD	−1.85	−3.03; −1.46	+3.52	+1.97; +3.94	+3.64	−2.41	−2.46; −0.07	−2.13	−1.44	−1.81; −0.84	+0.58	−0.30	−0.58; −0.02	+1.42	+1.07; +2.10	−3.93
BMI	SD	+1.46	+0.14; +2.22	+0.53	+0.07; +0.98	+2.91	−0.13	−1.37; +2.14	−0.33	+1.47	+1.18; +2.07	+0.07	+0.19	−0.66; +1.05	+0.95	−0.28; +0.99	−1.59
REE	Kcal/die	850	688; 920	1343	1290; 1451	1504	698	611; 1348	810	740	495; 1394	996	1078	945; 1211	892	846; 910	547
Prealb.	20–40 μg/dL	**19**	17; 20	**17**	16; 19	**16**	**15**	11; 18	**19**	**18**	16; 18	20	21	21; 21	**16**	13; 16	**18**
Leucine	75–107 μmol/L	**109**	97; 112	**110**	96; 121	**137**	**118**	117; 159	81	**123**	108.5; 126	**131**	**109**	109;109	90	82; 128	87
Insulin	2.6–25.0 mIU/L	9	4; 10.1	3.3	1.5; 5.2	7.7	3.3	3.3; 3.3	6.8	2.95	1.25; 4.15	**1.3**	7.65	7.2; 8.1	4.5	1.9; 4.5	7.7
Glyc.	70–110 mg/dL	76	70; 81	76	68; 81	85	85	75; 93	82	79	71; 82	62	77.5	63; 92	77	67; 77	80
HDL	>65 mg/dL	**49**	45; 54	**60**	53; 68	**54**	**55**	35; 58	**62**	**46**	46; 53	**54**	**50**	40; 60	**60**	47; 63	65
ApoA1	120–176 mg/dL	123	117; 128	127.5	122.0; 133.5	121	**112**	103; 131	121	**116**	104; 117	139	132.5	129;136	**118**	114; 135	144
ApoA2	25.1–34.5 mg/dL	25.9	23.2; 28.4	30.2	29.1; 32	31.3	**23**	18.4; 32	27.1	30	24.8; 30.5	**20.6**	**21.1**	21.1; 21.1	26.4	18.8; 26.5	31.4
ApoA	108–225 mg/dL	132	123; 140	155	133; 164	133	128	100; 150	148	128	124; 131	136	142	142; 142	134	118; 139	161
LDL	<130 mg/dL	89	74; 98	93.5	78.0; 110.0	104	77	47; 122	91	68	67; 81	66	104	75; 133	86	82; 90	64
ApoB	60–133 mg/dL	79	68; 84	69.5	63; 77	82	**57**	56; 109	86	69	67; 77	**56**	88.5	66.0; 111	74	60; 77	65

**Table 2 genes-16-00104-t002:** REE values of patients vs. controls paired according to age and sex (column ContA), height and sex (column ContB), with height > 1.88DS (column ContC) and Mann–Whitney U test results. Statistically significant values (* = *p* value ≤ 0.05, ** = *p* value ≤ 0.01, *** = *p* value ≤ 0.001) are reported in bold.

	N	Patients	ContA Median (Q1; Q3)	Patients vs. ContA	ContB Median (Q1; Q3)	Patients vs. ContB	ContC Median (Q1; Q3)	Patients vs. ContC
Group A	25	850 (688–996)	940 (733–1343)	0.112	864 (623–960)	0.54	-	-
RSTS1	9	850 (688–920)	1027 (864–1453)	0.053	733 (631–960)	0.67	-	-
Group B	10	1358 (1290–1455)	931 (817–1258)	**0.023 ***	822 (663–1158)	**0.001 *****	1091 (719–1255)	**0.0039 ****

**Table 3 genes-16-00104-t003:** RSTS1 and SS1 group: anthropometric measurements.Statistically significant values (* = *p* value ≤ 0.05, ** = *p* value ≤ 0.01, *** = *p* value ≤ 0.001) are reported in bold.

	RSTS (n = 11)		SS (n = 10)		RSTS vs. SS. *p*-Value
	Median	Q1; Q3	Median	Q1; Q3	
Weight (SD)	−0.02	−0.34; +1.70	+1.77	+0.62; +2.03	**0** **.016 ***
Height (SD)	−2.86	−3.79; −1.64	+2.01	+0.96; +2.99	**<0.0001 *****
OFC (SD)	−1.85	−3.03; −1.46	+3.52	+1.97; +3.94	**<0.0001 *****
BMI (SD)	+1.46	+0.14; +2.22	+0.53	+0.07; +0.98	0.118
REE (Kcal/die)	850	688; 920	1343	1290; 1451	**<0.0001 *****
Prealbumin (μg/dL)	19	17; 20	17	16; 19	0.093
Leucine (μmol/L)	109	97; 112	110	96; 121	0.706
Insulin (mIU/L)	9.0	4.0; 10.1	3.3	1.5; 5.2	**0** **.037**
Glycemia (mg/dL)	76	70; 81	76	68; 81	0.932
HDL (mg/dL)	49	45; 54	60	53; 68	**0** **.033**
ApoA1 (mg/dL)	123	117; 128	127.5	122.0; 133.5	0.340
ApoA2 (mg/dL)	25.9	23.2; 28.4	30.2	29.1; 32.0	**0** **.004 ****
ApoA (mg/dL)	132	123; 140	155	133; 164	0.137
LDL (mg/dL)	89	74; 98	93.5	78.0; 110.0	0.325
ApoB (mg/dL)	79	68; 84	69.5	63; 77	0.244

## Data Availability

The data presented in this study are available on request from the corresponding author due to ethical reasons.

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
