# Peer review of "The Epigenetic Machinery and Energy Expenditure: A Network to Be Revealed"

_genes, 2025, doi:10.3390/genes16010104_

Round 1

Reviewer 1 Report

Comments and Suggestions for Authors

This well-designed study investigates the metabolic foundations of growth abnormalities in Mendelian disorders of the epigenetic machinery (MDEMs), analyzing data from 38 patients with a focus on resting energy expenditure (REE) and metabolic parameters. The results demonstrate correlations between REE and physical measurements, while also revealing a notably higher REE in Sotos Syndrome type 1 patients compared to controls, independent of height.

Two suggestions:

1. Analyze potential gender correlations with REE associations to weight (A), height (B) and OFC (C)

2. Include participants' ethnicity data, or if not analyzed, address this limitation in the discussion as a relevant variable for future investigation.

Additionally, Replace Figure 2 with a higher resolution version.

Reviewer 2 Report

Comments and Suggestions for Authors

The introduction section does not clearly and comprehensively describe the research questions. Please reconsider the structure and flow to ensure that the purpose and significance of the study are articulated in a logical and detailed manner.

1. The introduction and abstract should clearly define what Mendelian disorders of the epigenetic machinery (MDEMs) are. Instead of simply stating “defects in these genes,” accurately describe MDEMs as genetic disorders involving disruptions in various components of the epigenetic machinery.

2. Please provide a clear rationale for studying MDEMs in the introduction. Is it due to their high prevalence or some other reasons? Strengthen the importance of investigating this disease.

3. Please specify the type of data collected for this study. Provide more details to give readers a clear understanding of the dataset.

4. When mentioning thermogenesis and physical activity, explain why only basal metabolic rate (BMR) was chosen for analysis. Justify its relevance and importance in the context of your study. 

5. The cohort appears to include different subtypes of MDEMs with varying age distributions and sample sizes. How were these factors controlled to ensure they did not bias your results or affect your conclusions?

Reviewer 3 Report

Comments and Suggestions for Authors

Comments to the authors:

I have carefully reviewed this manuscript, while the study provides insight into energy expenditure in Mendelian disorders of the epigenetic machinery (MDEMs), the novelty is not clearly articulated. The authors should better highlight how their findings advance understanding beyond existing literature, especially considering the niche topic of epigenetic disorders and metabolic syndromes. The paper seems narrowly focused and might struggle to engage a broader audience. A clearer exposition of the broader clinical implications (e.g., potential therapeutic interventions) could enhance its appeal. Some concerns that need to be addressed by the authors:

1: The study includes only 38 subjects, divided into small subgroups. This small sample size, while constrained by the rarity of MDEMs, severely limits the statistical power and generalizability of the findings. The authors should explicitly acknowledge this limitation in the abstract and conclusions.

2: Grouping by overgrowth (SS1 and WVS) and growth failure (RSTS1, CSS1, etc.) is not fully justified. Are there overlapping features in the syndromes that might blur these distinctions? More detailed criteria for subgrouping are needed.

3: The use of indirect calorimetry is appropriate, but the study lacks data on potential confounders like body composition (lean mass vs. fat mass) or activity levels, which could impact resting energy expenditure (REE).

4: Age- and sex-matched controls are mentioned, but it is unclear how many controls were included and whether they were appropriately matched for body composition or lifestyle.

5: Figure 2: The correlation plots lack sufficient annotation. For instance, indicating the statistical significance of trends directly on the plots would improve clarity. Table 1 and Table 3: The presentation of biochemical data is dense and difficult to interpret without further context. Highlighting significant deviations more effectively (e.g., bold font or separate columns) would help.

6: The paper relies heavily on non-parametric tests due to small sample sizes. However, these methods are not always ideal for drawing strong conclusions. The authors could consider bootstrapping techniques to improve robustness. The p-values are reported inconsistently (e.g., "p<0,05" vs. "p=0.023"). Standardize the notation and ensure appropriate significance thresholds.

Comments on the Quality of English Language

 The English could be improved to more clearly express the research.

Round 2

Reviewer 3 Report

Comments and Suggestions for Authors

The authors have addressed all the concerns